# A spatial interpolation method based on 3D-CNN for soil petroleum hydrocarbon pollution

**Sheng Miao**[1], **Guoqing Ni**[1], **Guangze Kong**[1], **Xiuhe Yuan**[2], **Chao Liu**[2]*, **Xiang Shen**[3], **Weijun Gao**[4]

1 School of Information and Control Engineering, Qingdao University of Technology, Qingdao, China,
2 School of Environment and Municipal Engineering, Qingdao University of Technology, Qingdao, China,
3 Department of Statistic, The George Washington University, Washington DC, United States of America,
4 Faculty of Environmental Engineering, The University of Kitakyushu, Kitakyushu, Japan

* liuchao@qut.edu.cn

**Data Availability Statement:** All files are available from the github database (at https://github.com/storydd/Spatial-Interpolation/tree/master).

**Funding:** The author(s) received no specific funding for this work.

## Abstract

Petroleum hydrocarbon pollution causes significant damage to soil, so accurate prediction and early intervention are crucial for sustainable soil management. However, traditional soil analysis methods often rely on statistical methods, which means they always rely on specific assumptions and are sensitive to outliers. Existing machine learning based methods convert features containing spatial information into one-dimensional vectors, resulting in the loss of some spatial features of the data. This study explores the application of Three-Dimensional Convolutional Neural Networks (3DCNN) in spatial interpolation to evaluate soil pollution. By introducing Channel Attention Mechanism (CAM), the model assigns different weights to auxiliary variables, improving the prediction accuracy of soil hydrocarbon content. We collected soil pollution data and validated the spatial distribution map generated using this method based on the drilling dataset. The results indicate that compared with traditional Kriging3D methods ($R^2 = 0.318$) and other machine learning methods such as support vector regression ($R^2 = 0.582$), the proposed 3DCNN based method can achieve better accuracy ($R^2 = 0.954$). This approach provides a sustainable tool for soil pollution management, supports decision-makers in developing effective remediation strategies, and promotes the sustainable development of spatial interpolation techniques in environmental science.

## Introduction

### Background

In the entire chain of exploration, extraction, transportation, processing, storage, and sales in the petroleum and petrochemical industry, poor management or accidents can lead to the leakage of petroleum pollutants, causing serious harm to the environment. When petroleum hydrocarbon pollutants enter the soil, they initially impact and harm the soil environment, affecting soil permeability, soil microbial diversity, and plant growth. Subsequently, these pollutants are absorbed by plants, affecting crop quality [1]. Those petroleum hydrocarbon pollutants not easily adsorbed by the soil can infiltrate underground with precipitation, polluting

**Competing interests:** The authors have declared that no competing interests exist.

shallow groundwater and affecting its quality [2]. Petroleum hydrocarbons in contaminated soil can enter the human or animal body through respiration, skin contact, oral ingestion, among other means. Prolonged exposure to substances like benzene, toluene, and phenols in petroleum hydrocarbons can lead to symptoms such as nausea, headache, and dizziness. Polycyclic aromatic hydrocarbons in petroleum hydrocarbons can disrupt the normal function of organs like the liver and kidneys, potentially causing cancer. Studies have shown that children frequently exposed to petroleum hydrocarbons face a four times higher risk of acute leukemia than average levels, with a seven times higher probability of acute non-lymphocytic leukemia compared to ordinary children. Therefore, accurate prediction of petroleum hydrocarbon content in soil, early treatment, and pollution prevention are crucial for soil environmental protection. Research data indicates that in vertically contaminated soil, petroleum hydrocarbons are mainly found in the surface layer (0-1.0m) and underground layer (1.0-2.0m), with lower concentrations in the remaining layers (2.0-3.0m, 3.0-4.0m, 4.0-5.0m), reflecting soil adsorption and retention. Concentrations decrease with increasing soil depth. Horizontally, there is uneven distribution, with higher soil petroleum hydrocarbon content near storage tanks, pump rooms, and oily wastewater treatment tanks [3].

Petroleum hydrocarbons, due to their inherent viscosity, tend to initially accumulate within the surface layer of soil upon entry. Through a combination of capillary action and gravity, these hydrocarbons gradually permeate downwards, eventually reaching saturation and migrating further into the groundwater under the influence of gravity, while also spreading horizontally within the soil. Once within the soil matrix, petroleum hydrocarbons undergo a process of degradation and transformation, which can be broadly classified into non-biological and biological pathways. Notably, photodegradation stands out as a significant non-biological degradation mechanism. Biodegradation emerges as the principal method for the conversion of petroleum hydrocarbons within the soil environment, facilitated by a diverse array of microorganisms, particularly those with lipophilic properties that specialize in breaking down both fatty and aromatic hydrocarbons. The effectiveness of petroleum hydrocarbon biodegradation is contingent upon several key factors, including the chemical characteristics of the hydrocarbons (bioavailability), the specific microbial metabolic pathways involved (with aerobic metabolism generally proving more rapid than anaerobic processes), and a range of environmental variables such as soil texture, pH levels, temperature, moisture content, salinity, oxygen availability, and nutrient concentrations [4]. Given the complex interplay of these factors, it is imperative to develop innovative spatial interpolation methodologies that leverage comprehensive soil environmental data and spatial information. Such approaches are vital for accurately depicting the distribution patterns of petroleum hydrocarbons within contaminated regions, thereby aiding in effective environmental management and mitigation strategies.

## Related works

Spatial interpolation is a pivotal method used to estimate values at unsampled points within two or three-dimensional space, enabling the creation of continuous surfaces or scenes across an entire spatial domain. Various techniques, including statistical-based interpolation, machine learning-based interpolation, polynomial interpolation, distance interpolation, and triangular mesh interpolation, are commonly employed for this purpose. The rise of neural networks has significantly enhanced spatial interpolation capabilities by enabling the interpolation of discrete data points and the seamless generation of surfaces through the nuanced learning of nonlinear data relationships. Neural networks, adaptable through diverse network architectures and training methods, offer tailored solutions for spatial interpolation tasks. Convolutional Neural Networks (CNNs) excel in processing grid and image data, while

Recurrent Neural Networks (RNNs) demonstrate effectiveness in handling sequential and spatiotemporal data. Moreover, advanced deep learning models like Generative Adversarial Networks (GANs) and Variational Autoencoders (VAEs) have found utility in spatial interpolation applications. The selection of an appropriate spatial interpolation method is critical for accurately predicting the spatial distribution of soil chemical elements within a region. Traditional deterministic and geostatistical interpolation methods, such as nearest neighbor interpolation, inverse distance weighted interpolation, kriging interpolation, trigonometric interpolation, and spline interpolation, have historically been utilized for soil attribute data interpolation. However, the neural networks has ushered in a new era, with these models increasingly shaping various interpolation fields, including image [5, 6], soil [7–9] and atmospheric sciences [10, 11], marking a significant advancement in the realm of spatial interpolation techniques.

Usually, traditional spatial interpolation methods rely on statistical analysis. These methods usually use statistical analysis of known data, so as to predict the target variable on the unmeasured location based on some known data. Classical methods include Kriging interpolation [12], Inverse Distance Weighting(IDW) [13] etc. These methods and their variants have been widely used in the field of spatial interpolation. [14] proposed an estimation calculation model of Kriging spatial interpolation method in the full spatial range of surrounding rock, and verified the applicability of the theoretical model. [15] use ordinary kriging method (OK) and general kriging method (UK) to predict the comprehensive surface water quality index (SWQI) of unobserved locations, and draw a SWQIs map of the study area to explore spatial distribution. [16] used Kriging interpolation technique to generate the soil shear wave velocity profile and evaluated its performance using field response analysis. [17] proposed a hybrid method combining three-dimensional kriging and depth function considering heterogeneity (3DK_DF) to improve the accuracy and reliability of three-dimensional spatial interpolation of soil heavy metals. In addition to commonly used spatial information, [18] introduces a new dual kriging extension called Spatiotemporal Dual Kriging (ST-DK), which establishes drift functions with fixed and adaptive coefficients. And experiments were conducted using temperature and pressure data from Thailand in 2018, and the results showed that when using adaptive coefficients, both ST-DK and ST-RK models outperformed fixed coefficient models. [19] uses a combination model of Random Forest (RF) and Ordinary Kriging (OK) to predict the spatial distribution of soil pollution using heavy metal data collected from abandoned metal mines. In summary, early kriging interpolation methods almost only used spatial relationships of interpolation attributes for interpolation, which is related to the distance between different position points. With the improvement of Kriging algorithm, in addition to spatial information, auxiliary information related to the target variable has been separately added. If the data points are too sparse or the spatial correlation is not obvious, Kriging interpolation may not be able to effectively estimate the attribute values of unknown locations. In addition, Kriging interpolation may not accurately reflect the true spatial variability, and Kriging interpolation is sensitive to outliers.

With the progress of technology, some scholars have noticed the use of location information and feature information for interpolation at the same time, and machine learning has attracted the attention of scholars. [20] employs deep learning algorithms for interpolating magnetic anomaly data, aiming to enhance the resolution of magnetic data. Additionally, gravity data are incorporated as supplementary information to improve the quality of magnetic anomaly data interpolation. [21] investigated the feasibility and effectiveness of three ML techniques, including the k-nearest neighbor algorithm, multigene genetic programming, and support vector regression (SVR), to estimate daily ET0 in Turkiye. Among the applied ML models, the SVR model provided the best results in determining ET0. [22] designed

 

hierarchical regionalization labels based on three interpolation techniques (inverse distance weight, ordinary kriging, and trend surface interpolation) as new spatial covariates for a machine learning (ML) model. It was demonstrated that regional spatial information improved the prediction performance of the model ($R^2 > 0.7$). [23] presents an innovative approach, known as the Discretized Spatial Encoding Neural Network (DSE-NN), comprising an encoder-decoder model designed on the basis of deep supervision, network visualization, and hyperparameter optimization. Through the discretization of input latitude and longitude data into specialized vectors, the DSE-NN adeptly captures temporal trends and augments the precision of reconstruction, concurrently addressing the complexity and fragmentation characteristic of oceanic data sets. [24] applies geostatistical methods (including ordinary kriging, regression kriging, and geographically weighted regression) and machine learning algorithms to generate high-resolution digital maps of surface soil characteristics in Romania. Zhang et.al [25] combined the Spatial AutoRegressive Neural Network (SARNN) with the GSDNN unit to construct the Generalized Spatial AutoRegressive Neural Network (GSARNN) for spatial interpolation in three-dimensional space. Machine learning interpolation methods are used for spatial interpolation. In contrast, 3DCNN can directly process 3D spatial data, such as volumetric data or stereoscopic images, and 3DCNN has local awareness and spatial context modeling capabilities. Through local receptive fields, convolutional layers can capture local spatial patterns and features in the data. At the same time, by stacking multiple convolutional layers, 3DCNN can learn a wider range of spatial context information and improve the spatial continuity and accuracy of the interpolation results. This gives 3DCNNs an advantage when dealing with data with complex spatial variability, such as data in geological attributes or environmental monitoring.

CNNS are a suitable and efficient method for processing images, video, and sound. Among them, 1D CNN is mainly applied to the processing of sequence data and time series data, 2D CNN is mainly applied to image data processing and computer vision tasks, and 3DCNN is mainly applied to video data and 3D data processing. 3DCNN convolutions data in space and time dimensions, which can capture the spatio-temporal relationship in data. Inspired by the similarity of video analysis, [26] propose a new pure spatio-temporal model based on 3D convolutional neural network (3DCNN) to simultaneously capture spatiotemporal features from low-level to high-level layers, and design a grouped 3D multiscale residual strategy to directly and effectively extract multiscale spatial features. [27] studied a wind prediction model based on deep learning. The proposed prediction model consists of three-dimensional convolutional neural network and deep Convolutional long Short-Term memory (3DCNN-DConvLSTM) to predict wind vectors in the form of time series images. The DConvLSTM model learns spatio-temporal features from time-series image data that represent spatio-temporal variations in wind speed and direction. [28] adopted the early fusion method of autocorrelation analysis for importance sampling, used a 3D multi-scale extended convolutional network to capture both near and long range correlations, and used a densely connected network for deeper feature extraction, and designed a new module called "Spatial and Channel Recalibilization" (SCR). [29] develops a 3-D star neural network (StarNet) for polarimetric radar QPEs that integrate physical height prior knowledge and employ a reweighted loss function. [30] analyzed the spatio-temporal variation characteristics of chub mackerel catch and fishing grounds in the northwest Pacific Ocean based on seven years of fishery statistics and Marine remote sensing data by applying the center of gravity of fisheries, 2DCNN and 3DCNN models. [31] proposed a 3D Convolutional Neural Network (3D CNN) model to nowcast a brief local convective storm event based on the unique 3D observation data of multi-parameter phased Array weather radar (MP-PAWR).

## Research gap and contributions

In practical applications, spatial interpolation should select the most suitable method based on the nature of the data and the specific requirements of the analysis, and the accuracy and reliability of the results obtained through spatial interpolation must be evaluated and verified using available independent data. Traditional interpolation methods typically only consider spatial correlation and often overlook the unique features of interpolated data. For example, in soil interpolation methods, there is relatively little research on methods that integrate spatial information and feature data related to soil physical and chemical properties. In addition, during the feature extraction process, machine learning methods often convert data into two-dimensional or one-dimensional formats, which may result in the loss of spatial information. This study uses a 3DCNN model to address these issues by utilizing spatial data and the physical and chemical properties of soil. This method is particularly advantageous for spatial data containing multiple environmental factors or geographical attributes. In addition, 3DCNN models are highly suitable for processing data with three-dimensional spatial structures.

This study aims to explore the application of a 3DCNN-based spatial interpolation method in soil pollution assessment and compare its accuracy and applicability. This study have collected existing soil pollution data and generated soil pollution distribution maps for the whole study area using the chosen spatial interpolation method. The interpolation results have also been verified and corrected to improve the understanding of the soil pollution distribution and the accuracy of the assessment. Through the results of this study, this paper expect to provide reliable tools and methods for soil pollution research and management to support policy makers in developing effective soil pollution management and remediation strategies. At the same time, this paper also hope to promote the further application and development of spatial interpolation techniques in the field of environmental science.

The main contributions are summarized as follows:

1. This paper proposes using a 3DCNN model to combine soil environmental factors and spatial information for spatial interpolation. In addition, this paper has processed the data into a three-dimensional matrix suitable for 3DCNN models, thereby promoting the utilization of spatial information.

2. This paer combines CAM with 3DCNN to assign different weights to auxiliary variables such as soil environmental factors and spatial information, thereby motivating the allocation of different weights to each channel (soil environmental factors and spatial information).

3. In this study, the effectiveness of the proposed 3DCNN interpolation model was validated based on a drilling dataset of soil petroleum hydrocarbon content.

The remaining parts of this paper are organized as follows. In Section 2, this paper summarize recent research related to this study. Section 3 describes the proposed approach, and in Section 4 we describe the experiments and results of the related dataset collection and preparation. Finally, Section 5 concludes the paper.

## Methodology

As shown in Fig 1, the research area covers an area of 11600 $m_2$ and is located in the Qingdao area of northern China, including living and office areas of coking plants and chemical enterprises. For the past 15 years, it has mainly produced MBS resin, with production, storage, and office areas. After 2022, all buildings and production facilities in the plot will be demolished. The terrain of the study area is flat, with a ground elevation of 4.65-5.28m.

The region is located in a coastal hilly area and belongs to a temperate marine climate. The dominant wind direction in the region is northwest wind, with an average annual precipitation

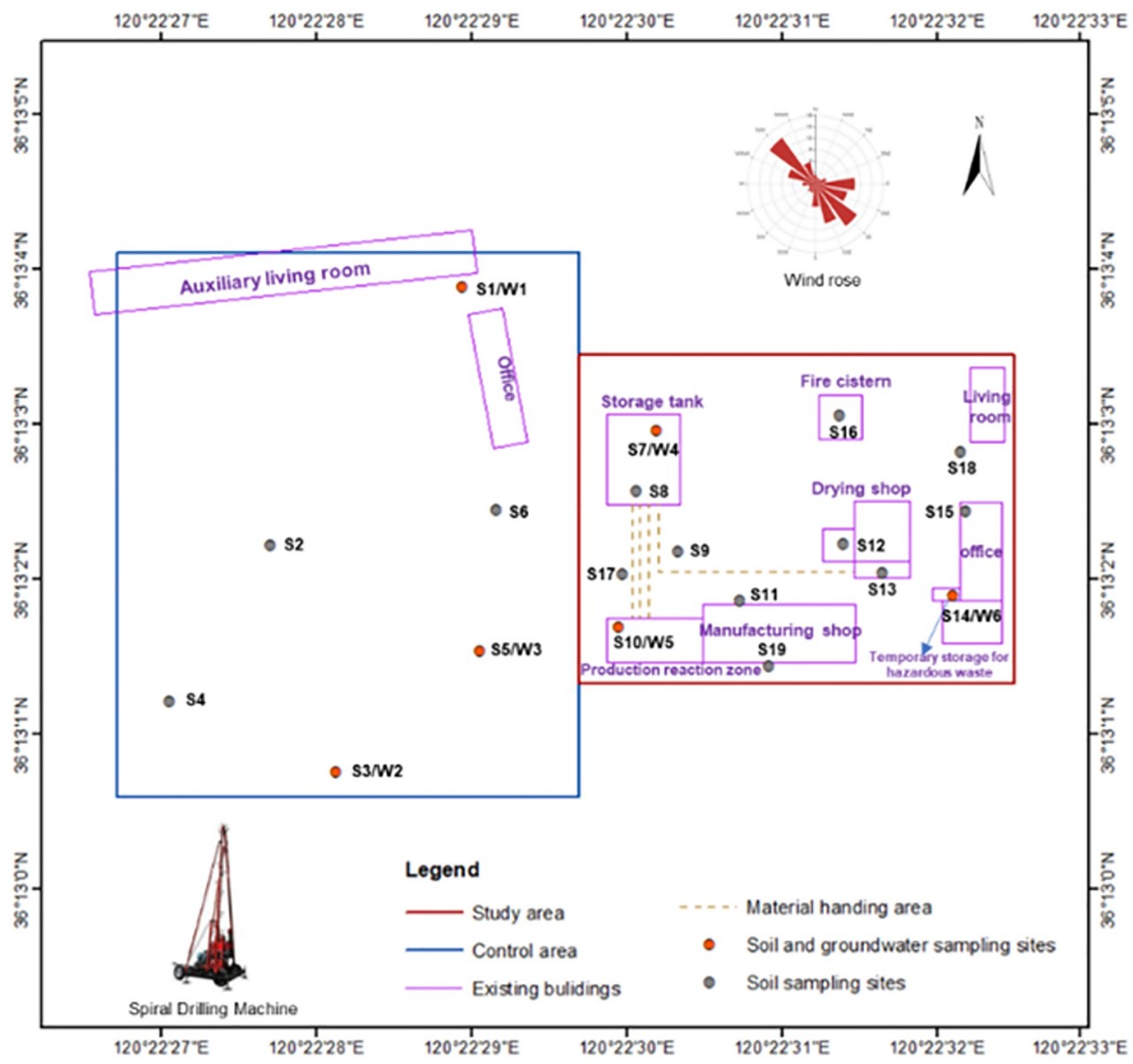

**Fig 1. Research area location.**

of about 755.6mm, an average annual temperature of 12.2°C, and an average wind speed of 3.9m/s. Within a radius of 1 kilometer outside the research area, there have been multiple types of enterprises in history, including coking plants, automobile manufacturing plants, chemical fiber material factories, machining enterprises, and anti-corrosion material factories, all of which have been operating for more than 10 years. The original production heat source of chemical enterprises is provided by diesel boilers. The possible pathways of petroleum hydrocarbon pollution in the research area include atmospheric sedimentation, leakage and infiltration, and groundwater migration. As shown in the box plot in Fig 2, the petroleum hydrocarbon content in some chemical enterprises is relatively high.

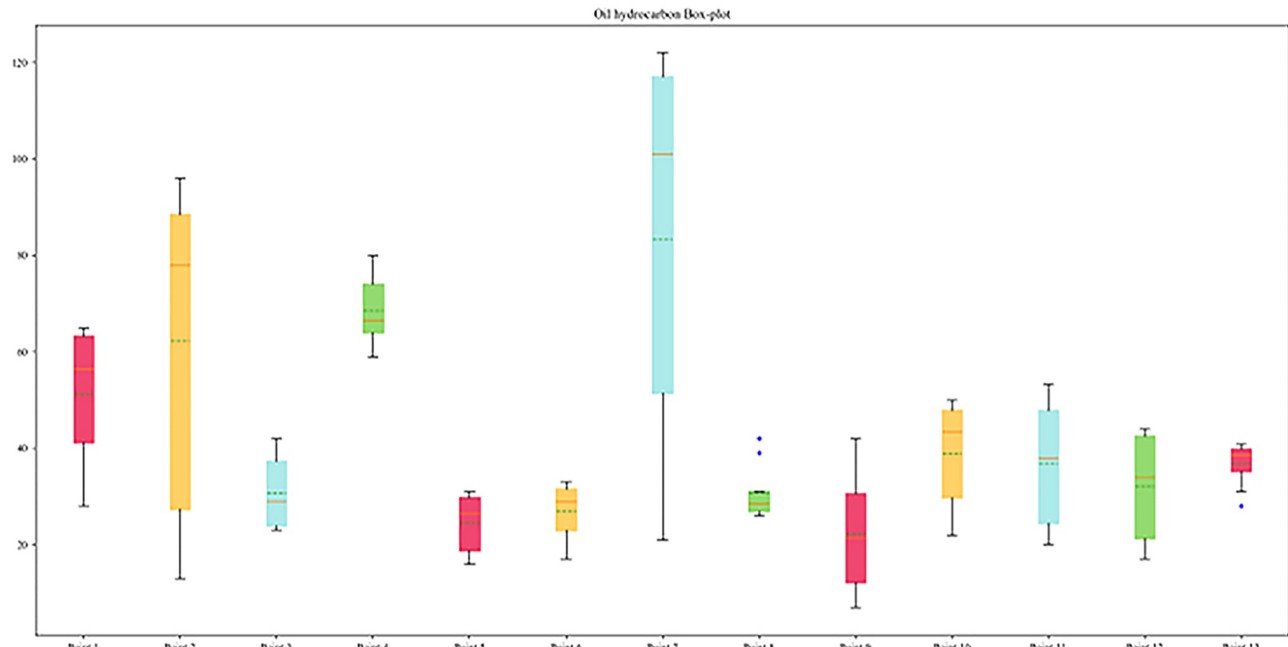

**Fig 2. Frequency distribution of soil petroleum hydrocarbons at sampling points.**

## Data preparing

In this study, soil boreholes were drilled in the study area, and the longitude, latitude, and depth of the borehole locations were recorded while collecting soil samples. The data collection for this field survey is authorized by the Qingdao Municipal Bureau of Ecology and Environment, China, and the results of the field survey are one of the necessary conditions for the secondary development of the site. The collected soil samples were tested to obtain soil physical and chemical properties (including pH value, wet bulk density, vertical permeability coefficient, organic carbon, average water supply, salinity) and petroleum hydrocarbon index. A total of 143 soil data were collected.

Considering that the range of different data in the data set is quite different, for example, the range scale of location information is 1000000, and the range scale of average water supply is 0.01, in order to eliminate the impact of dimension and data value range. This study uses Min Max normalization, which normalizes the data by assigning the mean and standard deviation to the original data. The processed data range is [0, 1], and its conversion formula is as Eq (1):

$$x' = \frac{x - x_{min}}{x_{max} - x_{min}} \tag{1}$$

Where $x_{min}$ is the minimum value of the original data, and $x_{max}$ is the maximum value of the original data, the distribution of each feature for each input is shown in Table 1.

## Modeling

3D Convolutional Neural Network (3DCNN) is mainly used in fields such as video classification and action recognition, and is an extension of 2D Convolutional Neural Network. Although 2DCNNs are good at capturing spatial information from images, they are difficult to

**Table 1. Data distribution.**

|  | Max | Min | Mean | Variance |
|---|---|---|---|---|
| Logitude(transformed) | 4009743.679 | 4009693.891 | 4009718.790 | 14.697 |
| Dimension(transformed) | 40533772.816 | 40533717.114 | 40533743.241 | 20.618 |
| Depth | 6.000 | 0.000 | 3.215 | 1.810 |
| PH | 7.740 | 7.610 | 7.673 | 0.045 |
| Wet Bulk Density | 1.620 | 1.430 | 1.493 | 0.058 |
| Permeability Coefficient | 15.000 | 0.020 | 10.406 | 4.644 |
| Organic | 0.830 | 0.170 | 0.450 | 0.198 |
| Average Water Yield | 0.270 | 0.020 | 0.215 | 0.084 |
| Salinity | 0.400 | 0.100 | 0.275 | 0.148 |

effectively capture temporal information. In contrast, 3DCNN performs convolution not only in the spatial dimension but also in the temporal dimension, enabling it to extract temporal information from consecutive frame images in videos. In 2DCNN, the filter generates a two-dimensional feature map as output, compressing information from multiple stacked images. However, in 3DCNN, the output is still a three-dimensional feature map. This means that when using 2DCNN for video processing, each frame is typically processed independently by the CNN model. This method does not take into account the inter frame motion information in the time dimension. On the other hand, by using 3DCNN, spatial and temporal features can be captured to more comprehensively represent videos. In summary, 3DCNN is specifically designed for processing video data by combining convolution in the temporal dimension. This enables them to effectively capture spatial and temporal information, thereby improving video analysis tasks such as action recognition and video classification.

The CAM the attention mechanism through the relationship between the internal features. The idea flow of channel attention is as follows: Firstly, in order to facilitate the later learning of channel features, pooling is carried out in the spatial dimension, and the spatial size is compressed. The global maximum pooling and global average pooling of the spatial dimension are performed on an input feature map F of size $H \times W \times C$, and two $1 \times 1 \times C$ feature maps are obtained. Then, the results of global Max pooling and global average pooling were sent to a shared Multi-Layer Perceptron (MLP) to learn respectively, and two $1 \times 1 \times C$ feature maps were obtained. Finally, Add operation was performed on the output result of MLP, and then the channel attention weight matrix $M_c \in R^{1 \times 1 \times C}$ was obtained after the mapping processing of Sigmoid activation function, and the process was shown in Fig 3.

This study proposes a petroleum hydrocarbon index interpolation network based on the fusion of 3DCNN and channel attention mechanism. It fully integrates the advantages of two mechanisms, improves the overall efficiency of the network, and the CAM gives different feature weights through training. 3DCNN fully utilizes the spatial characteristics of data and solves the problems in current soil interpolation recognition. This interpolation model mainly uses 3DCNN layers for multi-level spatial feature extraction, and trains the CAM part to give different weights to the channels in the feature maps extracted by 3DCNN. The proposed network architecture is shown in Fig 4.

The interpolation algorithm proposed in this paper is rooted in the 3D Convolutional Neural Network (3DCNN), with a comprehensive framework encompassing essential steps such as data preparation, data preprocessing, 3DCNN model construction, model training, model validation, interpolation processing, evaluation and optimization, and result analysis and display. Initially, crucial data preprocessing steps are executed to enhance the algorithm's stability. The

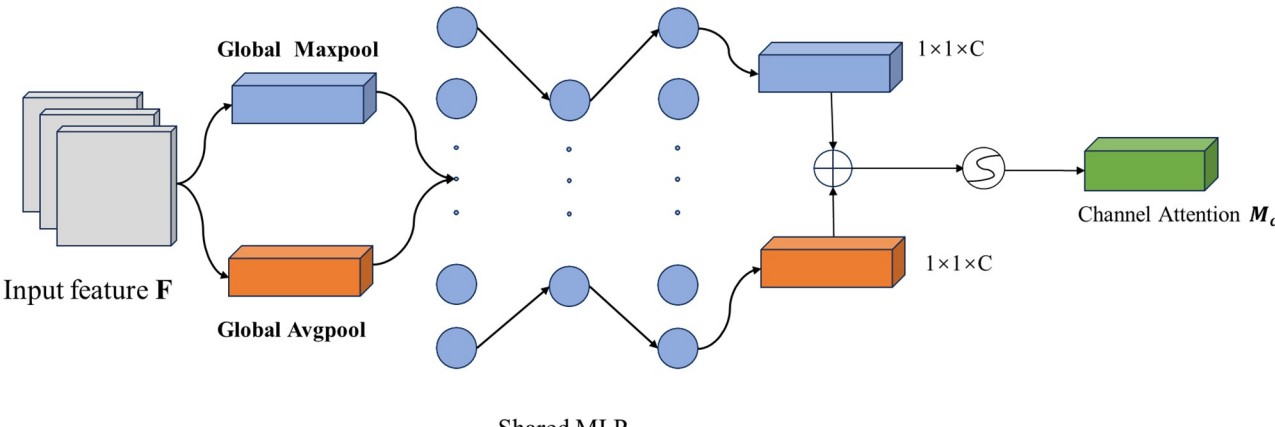

**Fig 3. Channel attention mechanism.**

feature data from neighboring points is organized into a matrix, with a suitable depth layer chosen as the third dimension for 3DCNN convolution. This structured data serves as both the input and output for the 3DCNN model. Following this, a 3DCNN model is crafted, comprising encoder and decoder components. The encoder facilitates data downsampling, while the decoder orchestrates the upsampling of the feature map to restore it to the original input data size.Subsequently, during the model training phase, a specific loss function and optimizer are defined to efficiently train the model. The model validation phase assesses the model's performance using a dedicated validation set. In the interpolation processing stage, the trained model is leveraged to interpolate new data, showcasing the model's capability in extrapolating information across the spatial domain.

This network consists of two 3DCNN layers for feature extraction of input feature matrices. The first two 3DCNN layers are followed by CAM mechanism to give channel weights to the extracted feature maps. Finally, two Conv3DTranspose layers are used for upsampling to restore the feature maps back to their original scale and predict the petroleum hydrocarbon index of each input point. The overall train process of the proposed interpolation model is shown in Fig 5.

Spatial feature extraction: The interpolation problem is to predict the spatial petroleum hydrocarbon index $y \in R^{D \times m \times n}$ from the input spatial feature matrix $X_D^{S \times T} \in R^{D \times m \times n \times T}$, where $S = (m \times n) = 1, 2, \ldots, N$ (where m and n are the number of rows and columns in the two-

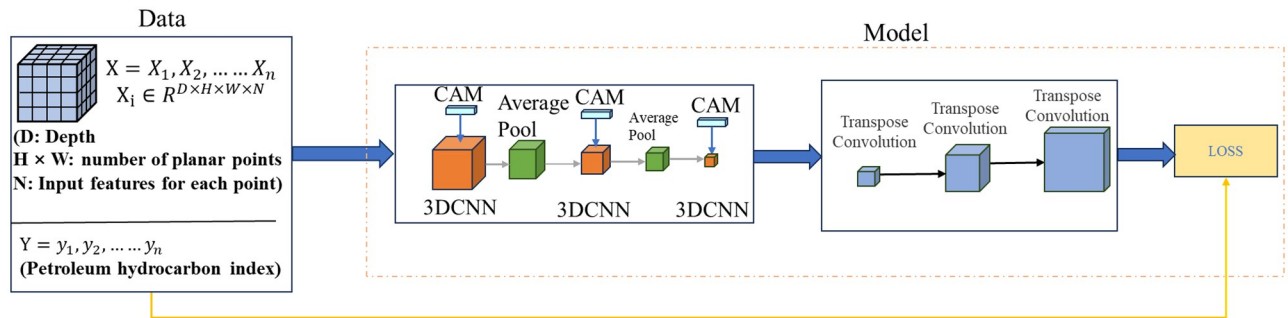

**Fig 4. Proposed 3DCNN architecture.**

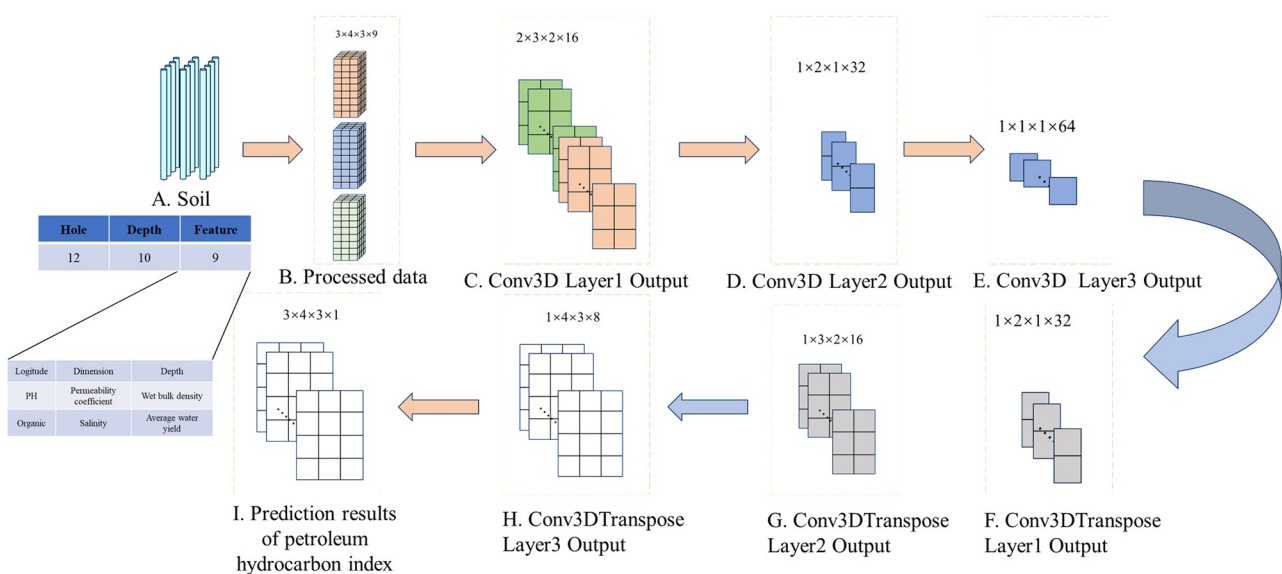

**Fig 5. Petroleum hydrocarbon array data flow layer-wise.**

dimensional matrix of soil boreholes, respectively) In this project, $m = 4$, $n = 3$, $D = 0 \sim D$ (D = different depths of soil boreholes) In this project, D = 3, T is the characteristic number of each two-dimensional soil spatial matrix. In this project, each point has 9 different characteristics, namely longitude, dimension, depth, pH, wet bulk density, permeability coefficient, unit organic carbon content, average water content, and salinity. For the proposed 3DCNN interpolation model, the spatial feature matrix X of soil boreholes is treated as an image of m x n size in a D frame, where each image has 9 channels as input, and y is the same size as the petroleum hydrocarbon index matrix and input spatial feature matrix for each point. Due to the interdependence of each soil borehole in the array in terms of latitude, longitude, and depth, CAM is used to assign different feature weights. Both feature maps and convolution kernels have depth parameters of 3DCNN, so convolution must also move in the depth direction. In the feature extraction stage of interpolation, the proposed model uses 3DCNN to extract features from the spatial matrix $[m \times n \times D]$. The input and output of each layer of the model are shown in Fig 5. The convolution operation and activation operation are merged into the convolution stage. The interpolation network structure consists of three layers from the input layer to the output layer. In addition to the input and output layers, there are also three 3DCNN layers, three pooling layers, and four Transpose convolution layers. The 3DCNN layer and pooling layer are alternately added, and batch normalization is performed after each layer to avoid overfitting. The Transfer layer is located behind the final pooling layer, and the 3D feature map extracted through feature extraction is upsampled and restored to a large output matrix of the same size as the input matrix through the Transfer layer. After each Transfer layer, batch normalization is used to avoid problems such as vanishing and exploding gradients, and the 'Tanh' activation function is nonlinearized to obtain the final output y. The output of the convolutional 3D layer can be calculated using the following formula:

$$P_{i,j}^{m,n,d} = Relu\left(b_{i,j} + \sum_{x=0}^{X_{i-1}}\sum_{y=0}^{Y_{i-1}}\sum_{z=0}^{Z_{i-1}}\omega_{ijk}^{xyz} X_{(i-1)k}^{(m+x)(n+y)(d+z)}\right) \tag{2}$$

Formally, the values of the position (m, n, d) on the feature map of the jth convolution kernel in the i-th layer are given by Eq (2), where $X_{i-1}$, $Y_{i-1}$, and $Z_{i-1}$ are the dimensions of the three-dimensional convolution kernel. X is the value of the k-th feature map in the previous layer, $\omega_{ijk}^{xyz}$ is the weight corresponding to the spatial coordinates of position ($p$, $q$, $r$) on the k-th feature map output in the previous convolutional or pooling layer, $b_{i,j}i$ is the bias constant, and $'Relu'$ is the activation function. The structure of the pooling layer is to minimize the amount of data and further reduce the spatial resolution of the input data to sample the input feature map, while expanding the receptive field. The pooling layer is used to process feature maps from multiple consecutive positions. The final upsampling layer uses four Conv3DTranspose layers, also known as deconvolution, to upsample the feature map to an interpolation method of the same size as the input matrix but with different spatial characteristics. The weight of the transposed convolution is learnable, which has a greater advantage in calculating the interpolation results of the petroleum hydrocarbon index matrix. A dropout processing method was used before Conv3DTranspose 1 layer to improve numerical performance and prevent overfitting. This study selected Mean Squared Error(MSE) as the loss function for model training, and Mean Absolute Error(MAE) is used as the indicator for detection, as shown in Eq (3).

$$MSE = \frac{1}{n}\sum_{i=1}^{n}(y_i - \hat{y})^2 \tag{3}$$

Among them, $y$ is the spatial matrix constructed from the actual petroleum hydrocarbon index at the input point, and $y^p$ is the output matrix of the proposed interpolation model.

The first convolutional layer of the proposed interpolation model has 16 filters with a kernel size of $2 \times 2 \times 2$ and a step size of 1. The outputs of 16 filters are shown in Fig 5C. The second and third convolutional layers contain 32 and 64 convolution kernels, respectively, with kernel sizes of $2 \times 2 \times 2$ and $1 \times 2 \times 1$, and a stride of 1. Similarly, the outputs of the second and third convolutional layers are given in Fig 5D and 5E. Meanwhile, after each convolution, the CAM is added to the feature map, assigning different weights to different channels. Use 3D transpose convolution to upsample the extracted feature maps. The first two Conv3DTranspose convolution kernels have the same 2D feature size as the corresponding 3DCNN used for feature extraction, namely $1 \times 2 \times 1$ and $1 \times 2 \times 2$. The convolution kernel size of the last Conv3DTranspose layer is $3 \times 1 \times 1$, which is used to regress each input point to obtain an output matrix with the same spatial dimension as the input spatial matrix. The value of each element is the petroleum index of that point, as shown in the Fig 5I.

## Experiment and result

This paper evaluates the performance of the prediction model using field collected datasets based on different evaluation indicators, and compares it with different spatial interpolation methods to verify the effectiveness of the proposed model. In the following section, this article has introduced the dataset used for training and evaluating the model, the results obtained, and conduct ablation experiments to demonstrate the effectiveness of the proposed model and the impact of the proposed components on the 3DCNN model.

### Dataset

The model proposed in this paper is trained and validated using soil drilling petroleum hydrocarbon data collected in the study area. The summarized data is arranged in coordinates, and each coordinate point has 10 different depth feature data and their corresponding petroleum

**Table 2. Hyperparameters of experiment.**

| Epoch | Learn Rate (delay) | CNN Layer Num | Activate Function | $R^2$ | MAE |
|---|---|---|---|---|---|
| 400 | 0.01(Y) | 3 | Tanh | 0.9547 | 0.0272 |
| 400 | 0.01(Y) | 2 | Tanh | 0.9042 | 0.0332 |
| 400 | 0.01(Y) | 2 | Relu | 0.9014 | 0.0361 |
| 400 | 0.01(Y) | 3 | Relu | 0.8781 | 0.0414 |
| 400 | 0.001(N) | 2 | Tanh | 0.9047 | 0.0385 |
| 400 | 0.001(Y) | 2 | Tanh | 0.7518 | 0.0625 |

hydrocarbon unit content. Use the minimum maximum normalization method to normalize the data to a range of 0-1, avoiding significant differences in the value range of different features that may affect model parameter updates. This study divided the total dataset into training, validation, and testing sets in a 7:2:1 ratio to train the proposed model.

## Experiment detail

Random search [32] identifies hyperparameters by quantitatively evaluating uniform random points in the hyperparameter array and selecting the point that produces the minimum MSE. This study conducts random search on the learning rate and the number of convolution kernels in each layer of 3DCNN, and uses R-squared($R^2$) as the model score. Additionally, the number of layers in the model is obtained through experiments, and the results of hyperparameter selection are shown in Table 2. In this study, the Adam optimizer was used to train the proposed model and the "Tanh" activation function was selected for all hidden layers. The proposed interpolation model uses the MSE loss function and a learning rate of 0.01. Meanwhile, using a learning rate decay strategy to detect validation set loss. When the loss function of the validator is not reduced, the learning rate decreases to 1/10 of the original value, avoiding the loss of obtaining local optimal solutions. The model is trained on a workstation, and the workstation configuration used in this study is shown in Table 3. Therefore, this study did not divide the data into batches for training. We selected a set of baseline spatial interpolation models for comparison, including Kriging 3D, SVR, and 3DCNN without CAM.

## Evaluation metrics

This study uses stochastic gradient descent optimization for parameter optimization and the 'Relu' activation function to nonlinearize neurons. Root Mean Squared Error(RMSE) represents the root mean square of the square difference between the actual and estimated petroleum hydrocarbon indices in the dataset. It is a measure of residual standard deviation. $R^2$ reports the proposal of variance in the dependent variable, as explained by the linear regression model. Therefore, when selecting RMSE and $R^2$ indicators based on the test set to compare the

**Table 3. Hyperparameters of experiment.**

| Setup | Specification |
|---|---|
| CPU | Intel Xeon Silver 4210 |
| GPU | RTX 3090 |
| Graphics memory | 24GB |
| Framework | Tensoflow2.6.0 |
| Code language | Python |

prediction accuracy of different interpolation models, the formulas are Eqs (4) and (5), MAE is calculated by taking the average of the absolute difference between predicted and actual values across the entire dataset. Compared with MSE, MAE does not decrease the error value due to the cancellation of positive and negative errors, so in some cases, MAE is considered a more appropriate error measurement indicator, the formulas are Eq (6).

$$RMSE = \sqrt{\frac{1}{n} \sum_{i=1}^{n} (y_i - \hat{y})^2} \tag{4}$$

$$R^2 = 1 - \frac{\sum_{i=1}^{n} (y_i - \hat{y})^2}{\sum_{i=1}^{n} (y_i - \bar{y})^2} \tag{5}$$

$$MAE = \frac{1}{n} \sum_{i=1}^{n} |y_i - \hat{y}| \tag{6}$$

Where, $y$ and $\bar{y}$ are the true and predicted labels for petroleum hydrocarbons. $\bar{y}$ is the average of $y$, and n is the amount of data in the test set.

## Results

The model has been trained and tested on petroleum hydrocarbon indices. In this study, ordinary 3DCNN and 3DCNN model combined with channel attention mechanism are used as interpolation models, and their training loss and test loss are shown in Figs 6 and 7, respectively. The MSE loss of the training set and the test set is reduced by 0.0001 and 0.0004, reaching less than 0.0017 and 0.0029. The observation in Fig 7 shows that the proposed method avoids the phenomena of overfitting and underfitting. Finally, the predicted indices of petroleum hydrocarbons are shown in Fig 8, which shows the true and predicted indices of petroleum hydrocarbons for 32 random samples in the test data.

The true petroleum hydrocarbon index is shown in blue and the predicted petroleum hydrocarbon index is shown in green. The actual indices and predicted concentrations of petroleum hydrocarbons are plotted in a similar manner. Fig 8 shows the actual and predicted indices of petroleum hydrocarbons for randomly selected samples from the test data, where blue is the true value and green is the predicted value. The evaluation metrics RMSE, $R^2$, and training time for different interpolation methods based on petroleum hydrocarbon datasets are shown in Table 4. The proposed soil petroleum hydrocarbon interpolation model was compared with other statistical based interpolation methods and deep learning based interpolation methods. The results of evaluation parameters show that this method has higher accuracy than other methods. However, in terms of training time, Kriging interpolation method and SVR are significantly shorter than 3DCNN, which needs to choose the appropriate method according to the needs of the experiment.

According to the data in Table 4, it can be seen that the CNN-CAM model proposed in this paper achieves the highest interpolation performance compared to the baseline model. This is because compared to Kriging3D, the model proposed in this paper can not only utilize spatial information features, but also use more soil physicochemical features, which are extracted through multiple convolutional layers to learn feature representations suitable for interpolation tasks. In addition, the proposed model outperforms the SVR model because 3DCNN can effectively capture contextual information in spatial data. It can recognize and extract spatial patterns and structures in data by performing convolution and pooling operations on three dimensions. In contrast, SVR is a statistical regression method that lacks the ability to directly handle spatial structures. By increasing the depth and width of the

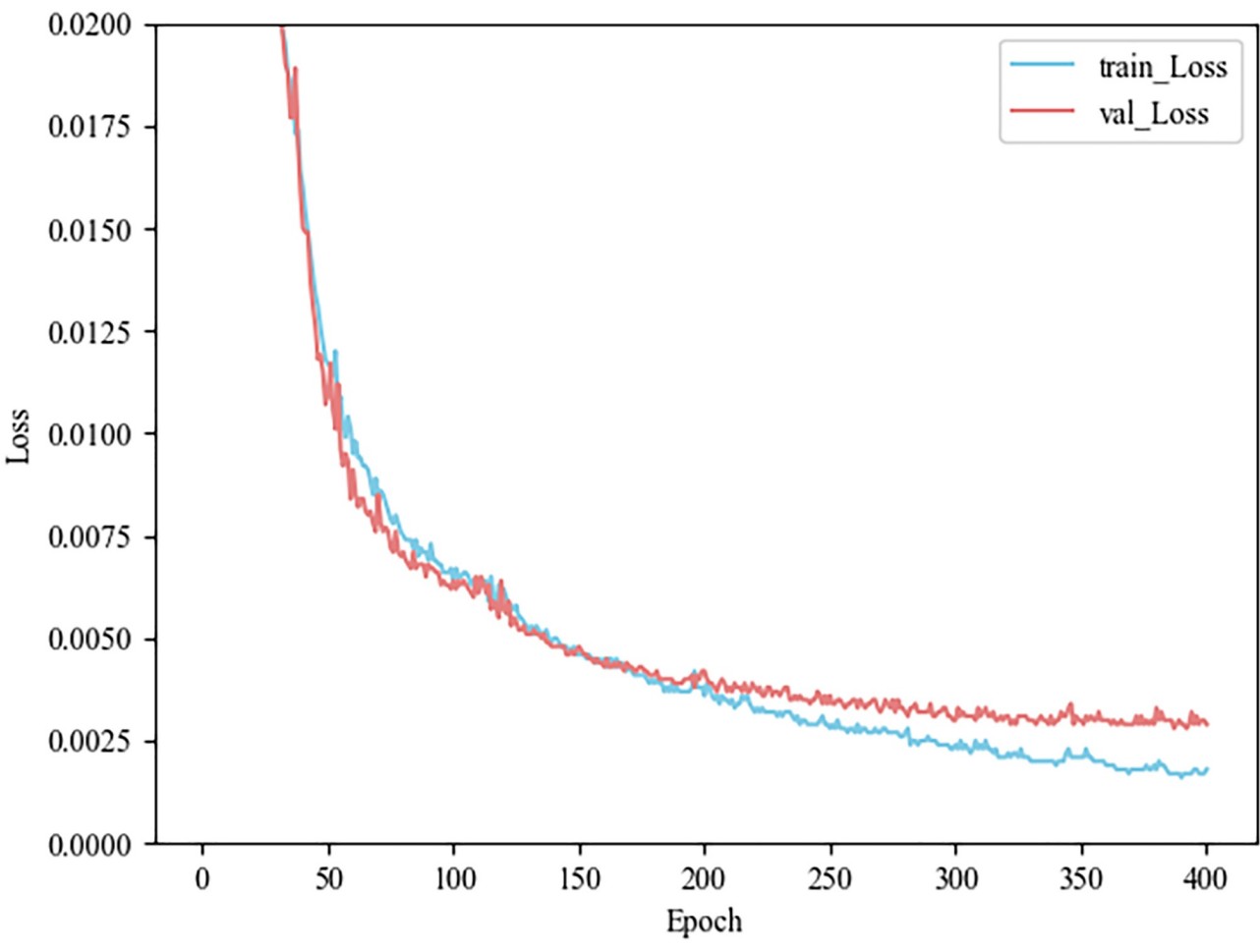

**Fig 6. Loss of 3DCNN.**

network, more complex spatial patterns and structures can be processed. In addition, 3DCNN can also expand training data and improve the generalization performance of the model through data augmentation techniques. SVR may face challenges such as dimensionality curse and model complexity when processing high-dimensional data. In summary, compared to SVR and Kriging3D, 3DCNN can better capture spatial contextual information and automatically learn feature representations when processing spatial data in spatial interpolation tasks. The CAM gives different features based on different weights, making the proposed model more focused on the most useful features for soil petroleum hydrocarbon interpolation tasks compared to a simple 3DCNN, improving the model's expressive power and performance. In summary, 3DCNN-CAM performs the best in spatial interpolation tasks.

## Ablation experiment

Here, this paper present an elaborate introduction to the ablation study examining the influence of the channel attention mechanism on the efficacy and runtime performance of the proposed interpolation technique. In each instance, this study modify a segment of the model at a

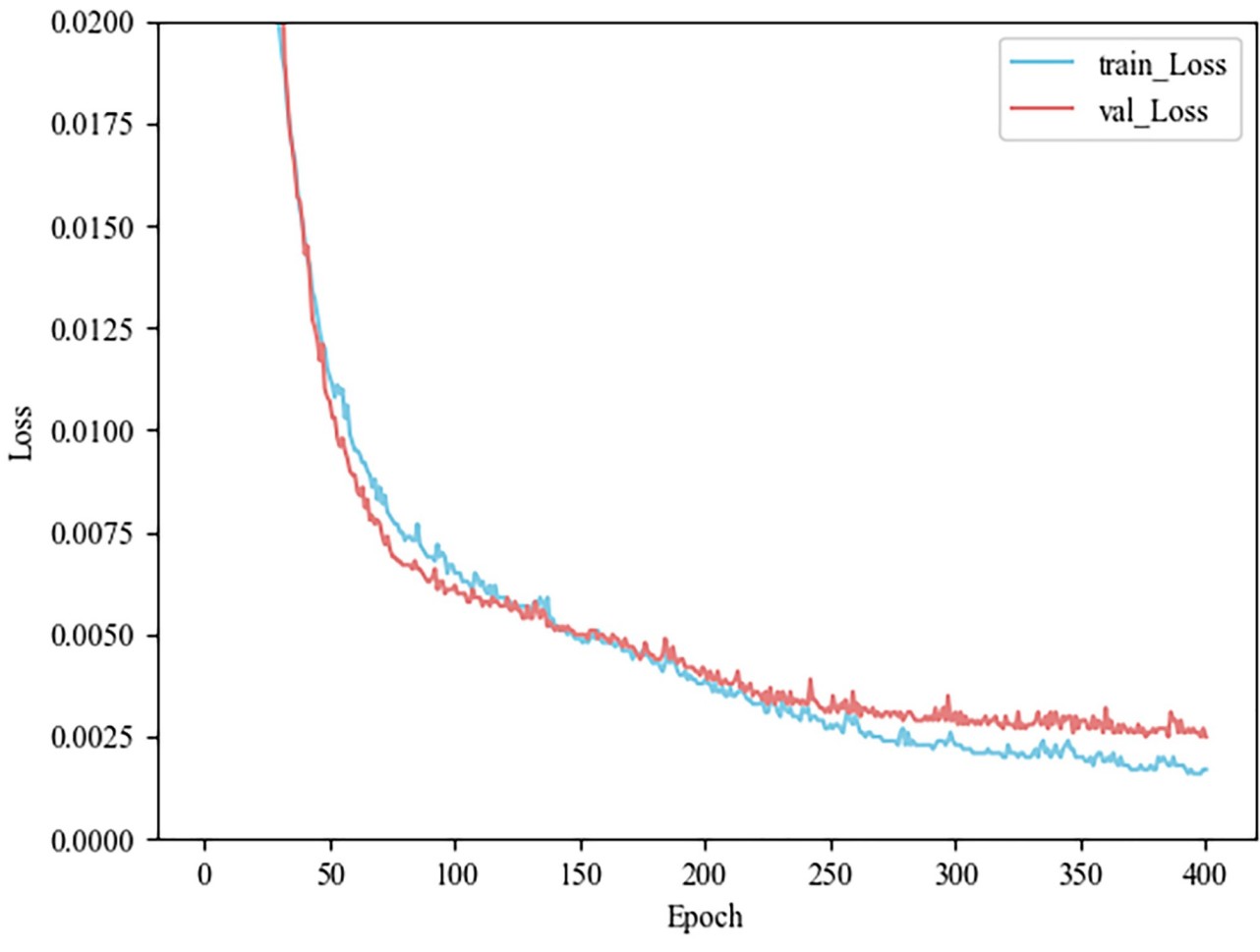

**Fig 7. Loss of 3DCNN with CAM.**

time, while maintaining the remaining CNN hyperparameters consistent with the optimal architecture for the specified dataset, as depicted in Table 5. Evidently, the 3DCNN outperforms the CNN model due to its utilization of three-dimensional spatial information, in contrast to the two-dimensional spatial information used by CNN. The model incorporating enhanced channel attention demonstrates superior performance because the primary function of the channel attention mechanism is to reveal the inter-channel correlations within the model. This mechanism autonomously discerns the significance of each feature channel (i.e., each environmental factor) through network learning, allocating distinct weight coefficients to individual channels. Consequently, this process fortifies critical features while dampening less crucial ones. Furthermore, this study conducted ablation experiments on features to verify environmental factor beyond spatial information and assessed the model's effectiveness. The results of these experiments are detailed in Table 6. Based on the insights gleaned from Table 6, the exclusion of attributes associated with wet bulk density, permeability coefficient, and organic matter content notably diminishes the model's performance. This outcome underscores the pivotal role of these three characteristics within the domain of soil environmental factors, particularly in the diffusion process of petroleum hydrocarbons.

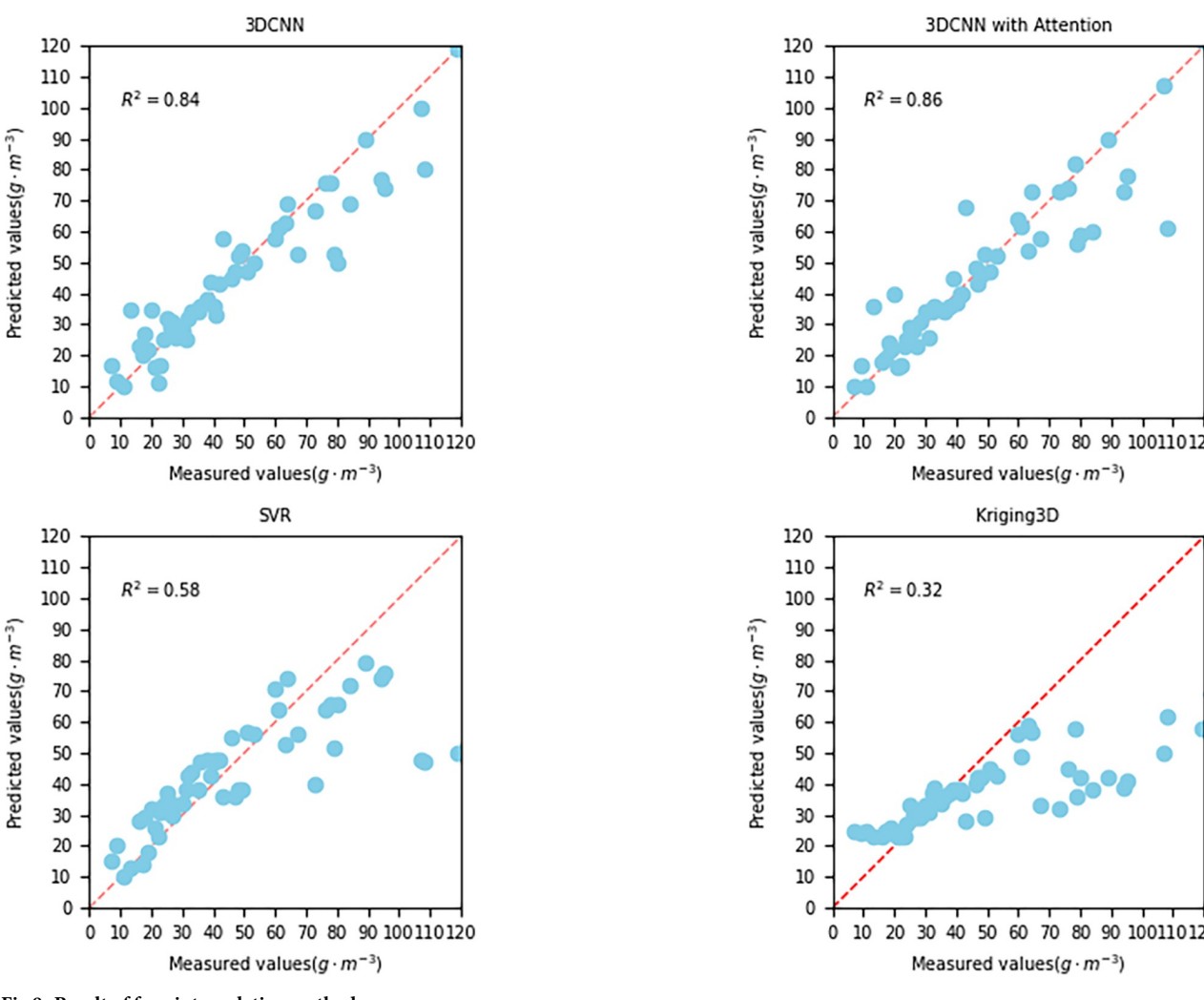

**Fig 8. Result of four interpolation method.**

## Discussion

The spatial distribution map of petroleum hydrocarbons in soil is shown in Fig 9: As can be seen from the figure, the high-value area of petroleum hydrocarbons mainly appears in the southeast direction of the study area, with a relatively large area. This area was originally a diesel boiler room, storage tank area, and production workshop reaction tank area. The formation

**Table 4. Result of 4 kinds of different spatial interpolation.**

| Method | $R^2$ | Rmse | MAE | Time |
|---|---|---|---|---|
| Kriging3D | 0.318 | 0.087 | 0.2240 | 0.01s |
| SVR | 0.582 | 0.256 | 0.1951 | 0.10s |
| FNN | 0.819 | 0.093 | 0.0748 | 7.1s |
| CNN | 0.801 | 0.080 | 0.0527 | 9.85s |
| Proposed model | **0.954** | **0.046** | **0.0272** | 11.65s |

**Table 5. Result of ablation experiment.**

| Method | $R^2$ | Rmse | MAE | Time |
|---|---|---|---|---|
| CNN | 0.801 | 0.080 | 0.0527 | 9.85s |
| 3DCNN | 0.839 | 0.086 | 0.0466 | 10.34s |
| Proposed model | **0.954** | **0.046** | **0.0272** | 11.65s |

**Table 6. Result of features ablation experiment.**

| Removed features | $R^2$ | Rmse | MAE |
|---|---|---|---|
| PH | 0.842 | 0.0797 | 0.0619 |
| Wet Bulk Density | 0.820 | 0.0852 | 0.0590 |
| Permeability Coefficient | 0.811 | 0.0874 | 0.0585 |
| Organic | 0.787 | 0.0928 | 0.0668 |
| Average Water Yield | 0.878 | 0.0700 | 0.0529 |
| Salinity | 0.866 | 0.0735 | 0.0534 |
| ALL Feature | **0.954** | **0.0460** | **0.0272** |

of this area may be mainly due to the leakage of crude oil during storage and transportation, as well as the leakage of crude oil or the infiltration of oily wastewater into the soil during some intermediate links of pipeline centralized transportation. Except for the high concentration of petroleum hydrocarbons in the soil mentioned above, other concentration values still have the characteristic of gradually decreasing from the pollution source to the peripheral areas.

## Conclusions

In this paper, a spatial interpolation method for the petroleum hydrocarbon index is proposed, which merges a 3DCNN deep learning model with a channel attention mechanism to attain optimal performance. The constructed model was validated utilizing a dataset on soil petroleum hydrocarbons. This approach delves into the intricate interplay between spatial

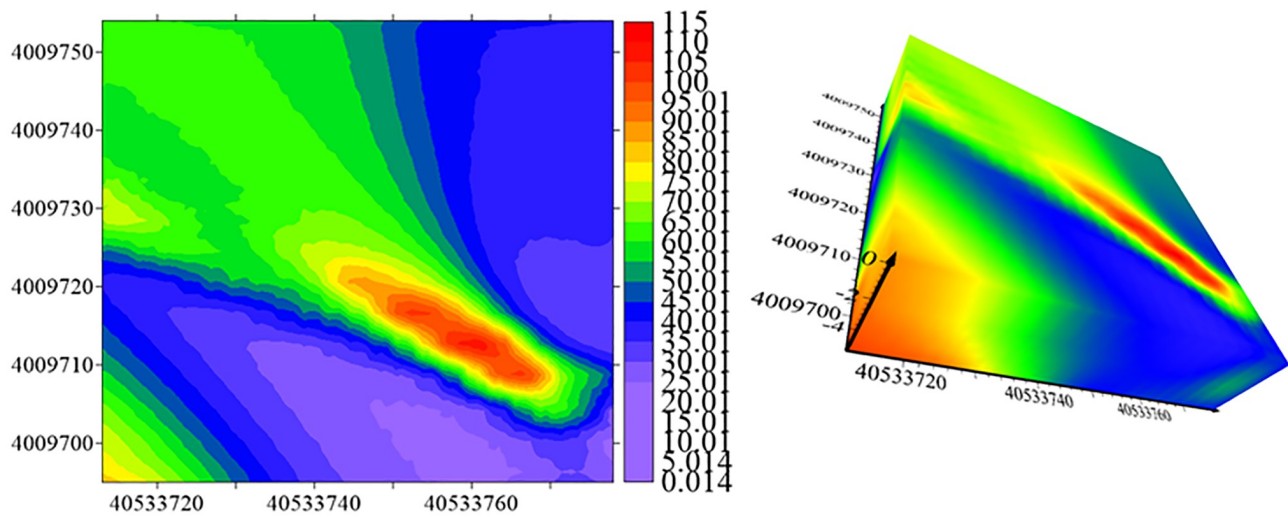

**Fig 9. Petroleum hydrocarbon prediction map using 3DCNN interpolation method.**

information and environmental factors (such as soil texture, pH value, temperature, moisture content, salinity, oxygen, and nutrients), resolving challenges encountered by traditional spatial interpolation methods, which often struggle to accurately estimate attribute values at unknown locations and are susceptible to outliers in cases of inconspicuous spatial correlation. In contrast to MLP, FNN, and SVR methods, the matrix structure of the data employed in this study proves more conducive for 3DCNN models, mitigating the limitations of numerous deep learning techniques that struggle to concurrently capture extensive spatial information. The integration of the 3DCNN deep learning model with the channel attention mechanism facilitates weight training across the dimensions of spatial information and environmental factors, enabling the assignment of distinct weights to each channel and surpassing the accuracy of the standalone 3DCNN model. By leveraging the correlation between spatial information and variables related to petroleum hydrocarbon distribution, this methodology achieves high-precision predictive models, demonstrating exceptional performance with an R2 value of 0.95 on unlabeled datasets. Moreover, this research identifies prospective applications where spatial interpolation utilizing pertinent auxiliary variables could be deployed in analogous regions.

Based on the outcomes of this study, the contribution involves furnishing dependable tools and methodologies for soil pollution investigation and management, thereby assisting policy makers in formulating effective strategies for soil pollution control and remediation. Simultaneously, the study aims to advance the application and evolution of spatial interpolation technology within the realm of environmental science. Despite the achievements of interpolation models in existing research, there remain challenges and opportunities for enhancement. The neural network theory underpinning the proposed interpolation model exhibits limited interpretability, posing a notable area for improvement within the model's performance. In forthcoming endeavors, it is advisable to delve into the mathematical principles underpinning traditional interpolation models and amalgamate them with novel models to enhance both the performance and interpretability of the interpolation model. This integrated approach holds promise for refining the efficacy and transparency of spatial interpolation models, thus fostering advancements in environmental science research and application.

## Acknowledgments

Cras egestas velit mauris, eu mollis turpis pellentesque sit amet. Interdum et malesuada fames ac ante ipsum primis in faucibus. Nam id pretium nisi. Sed ac quam id nisi malesuada congue. Sed interdum aliquet augue, at pellentesque quam rhoncus vitae.

## Author Contributions

**Conceptualization:** Chao Liu.

**Data curation:** Guangze Kong, Xiuhe Yuan.

**Formal analysis:** Weijun Gao.

**Methodology:** Sheng Miao.

**Software:** Guoqing Ni, Xiang Shen.

**Visualization:** Xiang Shen.

**Writing – original draft:** Guoqing Ni.

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
