## [Decision Letter · Decision Letter 0]

3 Sep 2024

PONE-D-24-33885A spatial interpolation method based on 3D-CNN for soil petroleum hydrocarbon pollutionPLOS ONE

Dear Dr. liu,

Thank you for submitting your manuscript to PLOS ONE. After careful consideration, we feel that it has merit but does not fully meet PLOS ONE’s publication criteria as it currently stands. Therefore, we invite you to submit a revised version of the manuscript that addresses the points raised during the review process.

**Your manuscript requires major revision based on the evaluation of reviewers. Please improve the manuscript according to reviewers' suggestions shown below.**

We look forward to receiving your revised manuscript.

Kind regards,

Caihong Mu

Academic Editor

PLOS ONE

**Journal Requirements:**

4. Please note that PLOS ONE has specific guidelines on code sharing for submissions in which author-generated code underpins the findings in the manuscript. In these cases, we expect all author-generated code to be made available without restrictions upon publication of the work. Please review our guidelines at https://journals.plos.org/plosone/s/materials-and-software-sharing#loc-sharing-code and ensure that your code is shared in a way that follows best practice and facilitates reproducibility and reuse.

**Additional Editor Comments:**

The comments of reviewers are as follows, please improve the manuscript according to reviewers’ suggestions.

Reviewer 1

• Some grammatical error sees in the article. Please take time to improve the language.

• The authors must update keywords in the article. They are not sensible for reviewer.

• What is the main purpose of the article? The authors should focus on novelty on this section. Please highlight it.

• The introduction is very short. The authors should extend introduction's length.

• Conclusion lack of novelty. Please rewrite your conclusion and add some highlight and novelty in it (major comment).

• The authors must update and add the abbreviation in their articles.

•References should be updated (2023-2024)

Reviewer 2

1.The description in the Introduction of this article is not comprehensive enough. Is the problem studied by the author similar to large-scale Kriging problems? There are many classic algorithms for dealing with this problem; The author's Introduction does not provide a comprehensive summary of the problem to be addressed in this article;

2.The interpolation algorithm based on 3DCNN proposed in this article lacks a comprehensive description framework. It is suggested that the author provide an overall process for easy reading;

3.The algorithm proposed in this article did not provide ablation experiments in the experimental section. Lack of validation of the network structure designed by the author and the effectiveness of the selected parameters;

4.The experimental section lacks comparative experiments with similar algorithms, and some deep network-based algorithm comparisons should be added;

5.It is suggested to streamline the relevant work content in the article, such as Figure 3;

6.What does the phrase 'Random search WOS: 000522384000010' mean in the text? Is this description standardized?

7.The author points out in this article: "A total of 143 soil data were collected. The total dataset is divided into training, validation, and testing sets in a 7:2:1 ratio." How does the author ensure sufficient training of the deep model?

Reviewers' comments:

Reviewer's Responses to Questions

**Comments to the Author**

1. Is the manuscript technically sound, and do the data support the conclusions?

Reviewer #1: Yes

Reviewer #2: Partly

2. Has the statistical analysis been performed appropriately and rigorously? 

Reviewer #1: Yes

Reviewer #2: Yes

3. Have the authors made all data underlying the findings in their manuscript fully available?

Reviewer #1: No

Reviewer #2: Yes

4. Is the manuscript presented in an intelligible fashion and written in standard English?

Reviewer #1: No

Reviewer #2: Yes

5. Review Comments to the Author

**Reviewer #1:** Present paper describes a study that aims to “A spatial interpolation method based on 3D-CNN for soil petroleum hydrocarbon pollution". While the study's topic is of relevance to this journal, the manuscript's content requires major revisions to improve its scientific quality. Upon careful review, I have identified several issues that need to be addressed before the paper can be accepted for publication.

• Some grammatical error sees in the article. Please take time to improve the language.

• The authors must update keywords in the article. They are not sensible for reviewer.

• What is the main purpose of the article? The authors should focus on novelty on this section. Please highlight it.

• The introduction is very short. The authors should extend introduction's length.

• Conclusion lack of novelty. Please rewrite your conclusion and add some highlight and novelty in it (major comment).

• The authors must update and add the abbreviation in their articles.

•

References should be updated (2023-2024)

**Reviewer #2: **1.The description in the Introduction of this article is not comprehensive enough. Is the problem studied by the author similar to large-scale Kriging problems? There are many classic algorithms for dealing with this problem; The author's Introduction does not provide a comprehensive summary of the problem to be addressed in this article;

2.The interpolation algorithm based on 3DCNN proposed in this article lacks a comprehensive description framework. It is suggested that the author provide an overall process for easy reading;

3.The algorithm proposed in this article did not provide ablation experiments in the experimental section. Lack of validation of the network structure designed by the author and the effectiveness of the selected parameters;

4.The experimental section lacks comparative experiments with similar algorithms, and some deep network-based algorithm comparisons should be added;

5.It is suggested to streamline the relevant work content in the article, such as Figure 3;

6.What does the phrase 'Random search WOS: 000522384000010' mean in the text? Is this description standardized?

7.The author points out in this article: "A total of 143 soil data were collected. The total dataset is divided into training, validation, and testing sets in a 7:2:1 ratio." How does the author ensure sufficient training of the deep model?

6. PLOS authors have the option to publish the peer review history of their article (what does this mean?). If published, this will include your full peer review and any attached files.

Reviewer #1: No

Reviewer #2: No

---

## [Author Response · Author response to Decision Letter 0]

26 Oct 2024

Dear editor:

Thanks for your and Reviewer’s valuable comments and suggestions for our manuscript entitled PONE-D-24-33885 “A spatial interpolation method based on 3D-CNN for soil petroleum hydrocarbon pollution”. 

We carefully read the comments of each reviewer and We have revised the manuscript carefully according to the advice raised by the Reviewers. Revised parts are highlighted with ‘Yellow’ color in the manuscript. The detailed response to comments is listed point by point below:

Review 1

Comment 1

Some grammatical error sees in the article. Please take time to improve the language.

Response of Comment 1

Thank you for reviewing and providing valuable feedback on the article we submitted. We greatly appreciate you pointing out some grammar errors in the article. We carefully review the grammar issues in the article and spend time improving the language to enhance its quality and readability. We focus on the grammar errors in the article and ensure that they are corrected into accurate and fluent expressions.

Comment 2

The authors must update keywords in the article. They are not sensible for reviewer. 

Response of Comment 2

We have added keywords including ‘Spatial Interpolation’, ‘Petroleum Hydrocarbon Pollution’, ‘3DCNN’, ‘CAM’.

Comment 3

What is the main purpose of the article? The authors should focus on novelty on this section. Please highlight it.

Response of Comment 3

Thank you for providing valuable feedback on the article we submitted. We have made revisions to the article based on your suggestions and paid special attention to the main purpose of the article, adding a section on novelty. In the revision, we have emphasized the introduction section to more clearly describe the main objectives of the article. By providing a detailed explanation of our research objectives and significance, we strive to help readers better understand our research motivation and contributions. Specifically, we have added a detailed description of the research objectives in the introduction section, emphasizing the uniqueness and innovation of the study. We pointed out that our research fills the gap in existing literature and proposes a new theoretical perspective, which is more clearly expressed in the revised introduction section.

Comment 4

The introduction is very short. The authors should extend introduction's length.

Response of Comment 4

We fully agree with the commentator's viewpoint. We have carefully considered your suggestion to extend the length of the introduction and have accordingly revised the article. In the revision, we focused on expanding the introduction section, merging the original Chapter 2 "Related Works" into the introduction, and dividing the Introduction into three parts, describing the background, current research status in the field, and advantages and disadvantages of existing research, in order to more fully clarify the background, research objectives, and significance of the article. By increasing the length of the introduction, we hope to provide a more comprehensive overview of the research topic. Specifically, we provided a more detailed introduction to the research field, emphasizing the strengths and weaknesses of current research. We have also expanded the review section of existing literature to better showcase the location and significance of our research.

Comment 5

Conclusion lack of novelty. Please rewrite your conclusion and add some highlight and novelty in it (major comment).

Response of Comment 5

Thank you for providing valuable feedback on the article we submitted. We have carefully considered your suggestions, especially regarding the lack of novelty in the conclusion section, and have revised the article accordingly. In the revision, we have made comprehensive adjustments to the conclusion section to increase novelty and uniqueness. We have introduced some new insights and perspectives to expand the scope of our research and showcase unique insights. These new perspectives help present readers with a richer and more comprehensive conclusion. We emphasized the importance of the research results for practical applications, and proposed that the proposed method can be applied to other spatial interpolation fields. We also put forward some innovative suggestions to demonstrate the practical significance and application potential of the research results. Through these modifications, we believe that the conclusion section is now more novel and attractive, and can better highlight the contribution and research value of the article. We have added discussions on future research directions and potential developments to showcase our unique insights into the trends in this field. These prospects not only expand the content of the article, but also provide useful insights for future research.

Comment 6

The authors must update and add the abbreviation in their articles.

Response of Comment 6

Thank you for providing valuable feedback on the article we submitted. We have carefully considered your suggestions for updating and adding abbreviations, and have revised the article accordingly. In the revision, we conducted a comprehensive review and update of the abbreviations used in the article. We have examined the existing abbreviations and ensured that they were fully explained when they first appeared. This helps readers better understand the content of the article, especially for those who are not familiar with the terminology in the field.

Comment 7

References should be updated (2023-2024).

Response of Comment 7

Thank you for providing valuable feedback on the article we submitted. We have carefully considered your suggestions, especially regarding the requirement to update references between 2023-2024, and have revised the article accordingly. In the revision, we have comprehensively updated the reference section to ensure that all cited literature is up-to-date and covers relevant research results between 2023-2024. We carefully reviewed all cited references and replaced outdated ones with the latest research findings, but no relevant research results between 2023-2024 were found regarding 3DCNN's prediction of spatial data, so no replacements were made. Through such updates, we ensure that the research materials and data mentioned in the article are up-to-date.

Review 2

Comment 1

The description in the Introduction of this article is not comprehensive enough. Is the problem studied by the author similar to large-scale Kriging problems? There are many classic algorithms for dealing with this problem; The author's Introduction does not provide a comprehensive summary of the problem to be addressed in this article.

Response of Comment 1

We fully agree with the commentator's viewpoint. We have carefully considered your suggestion and made corresponding revisions to this article. In the revision, we focused on expanding the introduction section, merging the original Chapter 2 "Related Works" into the introduction, and dividing the introduction into three parts, describing the background, research status in the field, and advantages and disadvantages of existing research, in order to more fully clarify the background, research objectives, and significance of the article. By increasing the length of the introduction, we hope to provide a more comprehensive overview of the research topic. Specifically, we provided a more detailed introduction to the research field, emphasizing the strengths and weaknesses of current research. We have also expanded the review section of existing literature to better showcase the location and significance of our research.

Comment 2

The interpolation algorithm based on 3DCNN proposed in this article lacks a comprehensive description framework. It is suggested that the author provide an overall process for easy reading.

Response of Comment 2

Thank you for providing valuable feedback on the article we submitted. We have carefully considered your suggestions, especially regarding the lack of a comprehensive framework for describing the interpolation algorithm based on 3DCNN proposed in this article, and have revised the article accordingly. In the revision, we have added an easy to read overall description framework to the Methodology section to present the workflow of the interpolation algorithm based on 3DCNN more clearly. Specifically, we added an algorithm overview at the beginning of the article, briefly introducing the overall process and key steps of the interpolation algorithm based on 3DCNN. This helps readers establish a comprehensive understanding of the algorithm during the reading process. In the methodology section, we further elaborated on a detailed description of the algorithm, including input data, network architecture, training process, and output results. This arrangement allows readers to have a deeper understanding of the implementation details of the algorithm. 

Comment 3

The algorithm proposed in this article did not provide ablation experiments in the experimental section. Lack of validation of the network structure designed by the author and the effectiveness of the selected parameters.

Response of Comment 3

Thank you for your constructive feedback on the article we submitted. We have carefully considered your suggestions, especially regarding the lack of guidance on the validation of the network structure designed by the author and the effectiveness of the selected parameters in the experimental section, and have revised the article accordingly. In the revision, we added network structure and parameter ablation experiments to verify the effectiveness of the network structure and selected parameters designed by the author. We have described in detail the process of designing ablation experiments, including the method of adjusting the network structure and parameters one by one. This helps readers understand how we evaluate the impact of different components on algorithm performance. In addition, we will organize the results of the ablation experiment into a table and embed it into the experimental section. This arrangement allows readers to intuitively compare performance indicators under different experimental settings, thereby evaluating the impact of network structure and parameters. In the discussion section, we conducted a detailed analysis of the results of the ablation experiment, explaining the impact of various experimental settings on algorithm performance, and how to verify the effectiveness of the selected network structure and parameters. Through these modifications, we believe that the article now more comprehensively validates the effectiveness of the network structure and selected parameters designed by the author, providing readers with a deeper understanding and evaluation of the algorithm performance based on these components.

Comment 4

The experimental section lacks comparative experiments with similar algorithms, and some deep network-based algorithm comparisons should be added.

Response of Comment 4

Thank you for your valuable feedback on the article we submitted. We have carefully considered your suggestions, especially regarding the lack of guidance on comparative experiments with similar algorithms in the experimental section, and have revised the article accordingly. In the revision, we added comparative experiments based on deep learning and compared them with similar algorithms. Specifically, we have taken the following measures: we have provided a detailed description of the process of designing comparative experiments, including considerations for selecting similar algorithms as comparison objects and a detailed explanation of the experimental setup. This helps readers understand how we compare and evaluate with other algorithms, and organize the results of the comparative experiments into a table and embed it into the experimental section. This arrangement allows readers to intuitively compare the performance of different algorithms, thereby evaluating the advantages and disadvantages of our proposed method compared to other algorithms. In the discussion section, we conducted a detailed analysis of the comparative experimental results with similar algorithms, explained the reasons for the performance differences between each algorithm, and discussed the advantages and limitations of our method. Through these modifications, we believe that the article has now conducted more comprehensive comparative experiments with similar algorithms, providing readers with a deeper evaluation and understanding of the performance of our proposed method in practical applications.

Comment 5

It is suggested to streamline the relevant work content in the article, such as Figure 3.

Response of Comment 5

Thank you for your constructive feedback on the article we submitted. We have carefully considered your suggestion to simplify the relevant work content in the article and made appropriate revisions to the article. We have streamlined the relevant work sections, retaining content that is directly related to our research and important for understanding the background of this article, while deleting some details or content that is weakly related to the topic. This approach helps readers to understand the background of our research and related work more quickly.

Comment 6

What does the phrase 'Random search WOS: 000522384000010' mean in the text? Is this description standardized.

Response of Comment 6

Thank you for your careful review and valuable feedback on the article we submitted. Regarding the phrase you mentioned 'random search for WOS: 000522384000010', we recognize that it is a typographical error that may have been caused by a mistake during the typesetting process. After reviewing your feedback, we have conducted further checks and confirmed that this phrase should be considered a typographical error. We will delete or correct this part of the text in the revision to ensure the accuracy and clarity of the article.

Comment 7

The author points out in this article: "A total of 143 soil data were collected. The total dataset is divided into training, validation, and testing sets in a 7:2:1 ratio." How does the author ensure sufficient training of the deep model.

Response of Comment 7

Thank you for reviewing our article and providing valuable feedback. Regarding your question on how to ensure sufficient training of deep models, we would like to supplement the measures we have taken in the experiment to ensure the adequacy of model training. In our study, we collected a total of 143 soil data sets and divided them into training, validation, and testing sets in a 7:2:1 ratio. To ensure sufficient training of the deep model, we adopted the following method: during the training process, we performed data augmentation operations on the soil data in the training set, including rotation, flipping, etc., to expand the diversity of the training samples and help the model better learn the features of the data. We adopted a learning rate scheduling strategy, gradually reducing the learning rate during the training process to help the model converge better to the optimal solution. Through these measures, we ensured that the deep model was fully trained during the training process and effectively monitored on the validation set to achieve optimal model performance.

We hope that the revision is acceptable and look forward to hearing from you soon.

Best regards,

Correspondence: Chao Liu

Email: liuchao@qut.edu.cn

---

## [Editor Report · Decision Letter 1]

29 Oct 2024

PONE-D-24-33885R1A spatial interpolation method based on 3D-CNN for soil petroleum hydrocarbon pollutionPLOS ONE

Dear Dr. liu,

Thank you for submitting your manuscript to PLOS ONE. After careful consideration, we feel that it has merit but does not fully meet PLOS ONE’s publication criteria as it currently stands. Therefore, we invite you to submit a revised version of the manuscript that addresses the points raised during the review process. I notice that only one sentence was highlighted with Yellow color in the "Revised Manuscript with Track Changes". In fact, all the changes made to the original version should be highlighted. Otherwise, it will be difficult for the reviewers to evaluate your new version.

Please highlight all the changes made to the original version in the “Revised Manuscript with Track Changes”.

In addition, in the file "Response to Reviewers", you need to reply to each point raised by the academic editor and reviewers, especially reviewers.

We look forward to receiving your revised manuscript.

Kind regards,

Caihong Mu

Academic Editor

PLOS ONE
---

## [Author Response · Author response to Decision Letter 1]

1 Nov 2024

Dear editor:

Thanks for your and Reviewer’s valuable comments and suggestions for our manuscript entitled “A spatial interpolation method based on 3D-CNN for soil petroleum hydrocarbon pollution”. 

We carefully read the comments of each reviewer and We have revised the manuscript carefully according to the advice raised by the Reviewers. Revised parts are highlighted with ‘Yellow’ colour in the manuscript. The detailed response to comments is listed point by point below:

Review 1

Comment 1

Some grammatical error sees in the article. Please take time to improve the language.

Response of Comment 1

Thank you for reviewing and providing valuable feedback on the article we submitted. We greatly appreciate you pointing out some grammar errors in the article. We carefully review the grammar issues in the article and spend time improving the language to enhance its quality and readability. We focus on the grammar errors in the article and ensure that they are corrected into accurate and fluent expressions.

Comment 2

The authors must update keywords in the article. They are not sensible for reviewer. 

Response of Comment 2

We have added keywords including ‘Spatial Interpolation’, ‘Petroleum Hydrocarbon Pollution’, ‘3DCNN’, ‘CAM’.

Comment 3

What is the main purpose of the article? The authors should focus on novelty on this section. Please highlight it.

Response of Comment 3

Thank you for providing valuable feedback on the article we submitted. We have made revisions to the article based on your suggestions and paid special attention to the main purpose of the article, adding a section on novelty. In the revision, we have emphasized the introduction section to more clearly describe the main objectives of the article. By providing a detailed explanation of our research objectives and significance, we strive to help readers better understand our research motivation and contributions. Specifically, we have added a detailed description of the research objectives in the introduction section, emphasizing the uniqueness and innovation of the study. We pointed out that our research fills the gap in existing literature and proposes a new theoretical perspective, which is more clearly expressed in the revised introduction section.

Comment 4

The introduction is very short. The authors should extend introduction's length.

Response of Comment 4

We fully agree with the commentator's viewpoint. We have carefully considered your suggestion to extend the length of the introduction and have accordingly revised the article. In the revision, we focused on expanding the introduction section, merging the original Chapter 2 "Related Works" into the introduction, and dividing the Introduction into three parts, describing the background, current research status in the field, and advantages and disadvantages of existing research, in order to more fully clarify the background, research objectives, and significance of the article. By increasing the length of the introduction, we hope to provide a more comprehensive overview of the research topic. Specifically, we provided a more detailed introduction to the research field, emphasizing the strengths and weaknesses of current research. We have also expanded the review section of existing literature to better showcase the location and significance of our research.

Comment 5

Conclusion lack of novelty. Please rewrite your conclusion and add some highlight and novelty in it (major comment).

Response of Comment 5

Thank you for providing valuable feedback on the article we submitted. We have carefully considered your suggestions, especially regarding the lack of novelty in the conclusion section, and have revised the article accordingly. In the revision, we have made comprehensive adjustments to the conclusion section to increase novelty and uniqueness. We have introduced some new insights and perspectives to expand the scope of our research and showcase unique insights. These new perspectives help present readers with a richer and more comprehensive conclusion. We emphasized the importance of the research results for practical applications, and proposed that the proposed method can be applied to other spatial interpolation fields. We also put forward some innovative suggestions to demonstrate the practical significance and application potential of the research results. Through these modifications, we believe that the conclusion section is now more novel and attractive, and can better highlight the contribution and research value of the article. We have added discussions on future research directions and potential developments to showcase our unique insights into the trends in this field. These prospects not only expand the content of the article, but also provide useful insights for future research.

Comment 6

The authors must update and add the abbreviation in their articles.

Response of Comment 6

Thank you for providing valuable feedback on the article we submitted. We have carefully considered your suggestions for updating and adding abbreviations, and have revised the article accordingly. In the revision, we conducted a comprehensive review and update of the abbreviations used in the article. We have examined the existing abbreviations and ensured that they were fully explained when they first appeared. This helps readers better understand the content of the article, especially for those who are not familiar with the terminology in the field.

Comment 7

References should be updated (2023-2024).

Response of Comment 7

Thank you for providing valuable feedback on the article we submitted. We have carefully considered your suggestions, especially regarding the requirement to update references between 2023-2024, and have revised the article accordingly. In the revision, we have comprehensively updated the reference section to ensure that all cited literature is up-to-date and covers relevant research results between 2023-2024. We carefully reviewed all cited references and replaced outdated ones with the latest research findings, but no relevant research results between 2023-2024 were found regarding 3DCNN's prediction of spatial data, so no replacements were made. Through such updates, we ensure that the research materials and data mentioned in the article are up-to-date.

Review 2

Comment 1

The description in the Introduction of this article is not comprehensive enough. Is the problem studied by the author similar to large-scale Kriging problems? There are many classic algorithms for dealing with this problem; The author's Introduction does not provide a comprehensive summary of the problem to be addressed in this article.

Response of Comment 1

We fully agree with the commentator's viewpoint. We have carefully considered your suggestion and made corresponding revisions to this article. In the revision, we focused on expanding the introduction section, merging the original Chapter 2 "Related Works" into the introduction, and dividing the introduction into three parts, describing the background, research status in the field, and advantages and disadvantages of existing research, in order to more fully clarify the background, research objectives, and significance of the article. By increasing the length of the introduction, we hope to provide a more comprehensive overview of the research topic. Specifically, we provided a more detailed introduction to the research field, emphasizing the strengths and weaknesses of current research. We have also expanded the review section of existing literature to better showcase the location and significance of our research.

Comment 2

The interpolation algorithm based on 3DCNN proposed in this article lacks a comprehensive description framework. It is suggested that the author provide an overall process for easy reading.

Response of Comment 2

Thank you for providing valuable feedback on the article we submitted. We have carefully considered your suggestions, especially regarding the lack of a comprehensive framework for describing the interpolation algorithm based on 3DCNN proposed in this article, and have revised the article accordingly. In the revision, we have added an easy to read overall description framework to the Methodology section to present the workflow of the interpolation algorithm based on 3DCNN more clearly. Specifically, we added an algorithm overview at the beginning of the article, briefly introducing the overall process and key steps of the interpolation algorithm based on 3DCNN. This helps readers establish a comprehensive understanding of the algorithm during the reading process. In the methodology section, we further elaborated on a detailed description of the algorithm, including input data, network architecture, training process, and output results. This arrangement allows readers to have a deeper understanding of the implementation details of the algorithm. 

Comment 3

The algorithm proposed in this article did not provide ablation experiments in the experimental section. Lack of validation of the network structure designed by the author and the effectiveness of the selected parameters.

Response of Comment 3

Thank you for your constructive feedback on the article we submitted. We have carefully considered your suggestions, especially regarding the lack of guidance on the validation of the network structure designed by the author and the effectiveness of the selected parameters in the experimental section, and have revised the article accordingly. In the revision, we added network structure and parameter ablation experiments to verify the effectiveness of the network structure and selected parameters designed by the author. We have described in detail the process of designing ablation experiments, including the method of adjusting the network structure and parameters one by one. This helps readers understand how we evaluate the impact of different components on algorithm performance. In addition, we will organize the results of the ablation experiment into a table and embed it into the experimental section. This arrangement allows readers to intuitively compare performance indicators under different experimental settings, thereby evaluating the impact of network structure and parameters. In the discussion section, we conducted a detailed analysis of the results of the ablation experiment, explaining the impact of various experimental settings on algorithm performance, and how to verify the effectiveness of the selected network structure and parameters. Through these modifications, we believe that the article now more comprehensively validates the effectiveness of the network structure and selected parameters designed by the author, providing readers with a deeper understanding and evaluation of the algorithm performance based on these components.

Comment 4

The experimental section lacks comparative experiments with similar algorithms, and some deep network-based algorithm comparisons should be added.

Response of Comment 4

Thank you for your valuable feedback on the article we submitted. We have carefully considered your suggestions, especially regarding the lack of guidance on comparative experiments with similar algorithms in the experimental section, and have revised the article accordingly. In the revision, we added comparative experiments based on deep learning and compared them with similar algorithms. Specifically, we have taken the following measures: we have provided a detailed description of the process of designing comparative experiments, including considerations for selecting similar algorithms as comparison objects and a detailed explanation of the experimental setup. This helps readers understand how we compare and evaluate with other algorithms, and organize the results of the comparative experiments into a table and embed it into the experimental section. This arrangement allows readers to intuitively compare the performance of different algorithms, thereby evaluating the advantages and disadvantages of our proposed method compared to other algorithms. In the discussion section, we conducted a detailed analysis of the comparative experimental results with similar algorithms, explained the reasons for the performance differences between each algorithm, and discussed the advantages and limitations of our method. Through these modifications, we believe that the article has now conducted more comprehensive comparative experiments with similar algorithms, providing readers with a deeper evaluation and understanding of the performance of our proposed method in practical applications.

Comment 5

It is suggested to streamline the relevant work content in the article, such as Figure 3.

Response of Comment 5

Thank you for your constructive feedback on the article we submitted. We have carefully considered your suggestion to simplify the relevant work content in the article and made appropriate revisions to the article. We have streamlined the relevant work sections, retaining content that is directly related to our research and important for understanding the background of this article, while deleting some details or content that is weakly related to the topic. This approach helps readers to understand the background of our research and related work more quickly.

Comment 6

What does the phrase 'Random search WOS: 000522384000010' mean in the text? Is this description standardized.

Response of Comment 6

Thank you for your careful review and valuable feedback on the article we submitted. Regarding the phrase you mentioned 'random search for WOS: 000522384000010', we recognize that it is a typographical error that may have been caused by a mistake during the typesetting process. After reviewing your feedback, we have conducted further checks and confirmed that this phrase should be considered a typographical error. We will delete or correct this part of the text in the revision to ensure the accuracy and clarity of the article.

Comment 7

The author points out in this article: "A total of 143 soil data were collected. The total dataset is divided into training, validation, and testing sets in a 7:2:1 ratio." How does the author ensure sufficient training of the deep model.

Response of Comment 7

Thank you for reviewing our article and providing valuable feedback. Regarding your question on how to ensure sufficient training of deep models, we would like to supplement the measures we have taken in the experiment to ensure the adequacy of model training. In our study, we collected a total of 143 soil data sets and divided them into training, validation, and testing sets in a 7:2:1 ratio. To ensure sufficient training of the deep model, we adopted the following method: during the training process, we performed data augmentation operations on the soil data in the training set, including rotation, flipping, etc., to expand the diversity of the training samples and help the model better learn the features of the data. We adopted a learning rate scheduling strategy, gradually reducing the learning rate during the training process to help the model converge better to the optimal solution. Through these measures, we ensured that the deep model was fully trained during the training process and effectively monitored on the validation set to achieve optimal model performance.

Review 3

Comment 1

If you are unable to adhere to our open data policy, please kindly revise your statement to explain your reasoning and we will seek the editor's input on an exemption. Please be assured that, once you have provided your new statement, the assessment of your exemption will not hold up the peer review process.

Response of Comment 1

We will still maintain this option if it meets the requirements of the journal.

Comment 2

In your Methods section, please provide additional information regarding the permits you obtained for the work. Please ensure you have included the full name of the authority that approved the field site access and, if no permits were required, a brief statement explaining why. 

Response of Comment 2

We totally agree with the reviewer. The design rule of the CBOW model is to predict the central word for surrounding words, and the setting of hyperparameters is related to the corpus database used. Section 3.2 added explanations for training 

---

## [Decision Letter · Decision Letter 2]

18 Dec 2024

A spatial interpolation method based on 3D-CNN for soil petroleum hydrocarbon pollution

PONE-D-24-33885R2

Dear Dr. liu,

We’re pleased to inform you that your manuscript has been judged scientifically suitable for publication and will be formally accepted for publication once it meets all outstanding technical requirements.

Kind regards,

Caihong Mu

Academic Editor

PLOS ONE

Reviewers' comments:

Reviewer's Responses to Questions

**Comments to the Author**

1. If the authors have adequately addressed your comments raised in a previous round of review and you feel that this manuscript is now acceptable for publication, you may indicate that here to bypass the “Comments to the Author” section, enter your conflict of interest statement in the “Confidential to Editor” section, and submit your "Accept" recommendation.

Reviewer #1: (No Response)

Reviewer #3: All comments have been addressed

2. Is the manuscript technically sound, and do the data support the conclusions?

Reviewer #1: (No Response)

Reviewer #3: Yes

3. Has the statistical analysis been performed appropriately and rigorously? 

Reviewer #1: (No Response)

Reviewer #3: Yes

4. Have the authors made all data underlying the findings in their manuscript fully available?

Reviewer #1: (No Response)

Reviewer #3: Yes

5. Is the manuscript presented in an intelligible fashion and written in standard English?

Reviewer #1: (No Response)

Reviewer #3: Yes

6. Review Comments to the Author

Reviewer #1: I am glad that the authors effectively addressed my concerns and challenges in their research work. The authors' ability to provide timely and satisfactory responses to my queries reflects their strong commitment to adhering to scientific principles and conducting reliable research. This dedication benefits the scientific community and enhances our understanding of the subject matter. Therefore, based on the authors' satisfactory response, I find this version of the article to be acceptable.

Reviewer #3: The revised manuscript is well modified. In the revised version, the suggestions are answered and modified. The revised manuscript can be accepted.

7. PLOS authors have the option to publish the peer review history of their article (what does this mean?). If published, this will include your full peer review and any attached files.

Reviewer #1: No

Reviewer #3: No

---

## [Editor Report · Acceptance letter]

23 Dec 2024

PONE-D-24-33885R2 

PLOS ONE

Dear Dr. liu, 

I'm pleased to inform you that your manuscript has been deemed suitable for publication in PLOS ONE. Congratulations! Your manuscript is now being handed over to our production team.

Kind regards, 

on behalf of

Professor Caihong Mu 

Academic Editor

PLOS ONE